# CV-YOLOv10-AR-M: Foreign Object Detection in Pu-Erh Tea Based on Five-Fold Cross-Validation

**DOI:** 10.3390/foods14101680

**Published:** 2025-05-09

**Authors:** Wenxia Yuan, Chunhua Yang, Xinghua Wang, Qiaomei Wang, Lijiao Chen, Man Zou, Zongpei Fan, Miao Zhou, Baijuan Wang

**Affiliations:** 1College of Tea Science, Yunnan Agricultural University, Kunming 650201, China; yuanwenxia2023@163.com (W.Y.); 2015056@ynau.edu.cn (L.C.);; 2Yunnan Organic Tea Industry Intelligent Engineering Research Center, Yunnan Agricultural University, Kunming 650201, China; 3College of Agronomy and Biotechnology, Zhejiang University, Hangzhou 310013, China

**Keywords:** CV-YOLOv10-AR-M network, MPDIoU, AssemFormer, Rectangular Self-Calibrated Module

## Abstract

To address the problem of detecting foreign bodies in Pu-erh tea, this study proposes an intelligent detection method based on an improved YOLOv10 network. By introducing the MPDIoU loss function, the YOLOv10 network is optimized to effectively enhance the positioning accuracy of the model in complex background and improve detection of small target foreign objects. Using AssemFormer to optimize the structure, the network’s ability to perceive small target foreign objects and its ability to process global information are improved. By introducing the Rectangular Self-Calibrated Module, the prediction accuracy of the bounding box is effectively optimized, further improving the classification and target-positioning abilities of the model in complex scenes. The results showed that the Box, Cls, and Dfl loss functions of the CV-YOLOv10-AR-M network in the One-to-Many Head task were, respectively, 14.60%, 19.74%, and 20.15% lower than those of the YOLOv10 network. In the One-to-One Head task, they decreased by 10.42%, 29.11%, and 20.15%, respectively. Compared with the original YOLOv10 network, the accuracy, recall rate, and mAP of the CV-YOLOv10-AR-M network were increased by 5.35%, 11.72% and 8.32%, respectively. The CV-YOLOv10-AR-M network effectively improves the model’s attention to small sizes, complex backgrounds, and detailed information, providing effective technical support for intelligent quality control in the agricultural field.

## 1. Introduction

As a local characteristic advantage industry, the Yunnan tea industry has become a key force in promoting regional economic growth by virtue of its significant industrial advantages and broad development prospects. As its core representative product, Pu-erh tea is widely favored by the market because of its unique flavor and profound cultural heritage. However, Pu-erh tea [1,2] is susceptible to contamination by various foreign materials during processing and storage, including bamboo leaves, pebbles, tree leaves, hair, and other impurities. These contaminants significantly compromise both the sensory quality and food safety of the final product. Conventional impurity detection methods primarily rely on manual sorting, which presents several limitations. First, the effectiveness of manual sorting is heavily dependent on the experience and concentration levels of workers. Additionally, this approach is prone to interference from factors such as operator fatigue, suboptimal lighting conditions, and subjective judgment errors. Furthermore, the low efficiency of manual sorting makes it inadequate for meeting the stringent requirements of modern high-yield production and quality control standards. Therefore, the intelligent detection technology with machine vision as the core has become a key factor in strengthening the quality control of Pu ’er tea and promoting the intelligent transformation of tea production.

With the widespread adoption of artificial intelligence [3] and of image processing technologies [4], machine vision has become one of the key technologies in agricultural perception, finding broad application and significantly promoting automation and intelligence in agricultural product processing. In recent years, researchers both domestically and internationally have achieved numerous representative results in applying machine vision technology to agriculture.

Xudong Sun et al. [5] proposed a detection method for tea stem impurities based on THz-TDS (Terahertz Time-Domain Spectroscopy) and imaging. Using both time-domain signals and frequency-domain absorption coefficients from THz-TDS, along with multiple baseline correction algorithms, the method enables rapid identification of tea stems. The AirPLS-KNN model constructed with the AirPLS algorithm was found to deliver the best performance, with accuracy of 97.3% and a recall rate of 96%, thus demonstrating high recognition precision and stability.

Zhan Shu [6] developed a hyperspectral imaging-based method for detecting visually similar foreign objects in chili peppers. A spectral pattern recognition algorithm was employed for pixel-level classification, with distinct colors used to highlight objects of the same category, thereby enhancing their visibility. Object detection algorithms were then applied for final localization. Experimental results showed that the Random Forest classifier achieved 86% accuracy, while YOLOv5 achieved accuracy of above 96%.

Ali Saeidan [7] introduced a hyperspectral imaging-based method for identifying foreign objects during cocoa bean processing, with the aim of improving product quality. Results showed that the SVM model outperformed other classifiers, achieving an overall accuracy of over 89.10%.

Although studies such as these have made considerable progress in improving recognition accuracy and model development, several challenges persist in detecting foreign objects in Pu-erh tea. Complex backgrounds, heterogeneous object types, and low-resolution [8] imaging can cause partial information loss, leading to missed detections or false positives. To address these issues, this study improves the YOLOv10 network [9,10] by incorporating the MPDIoU loss function [11,12], which enhances localization performance by directly considering the geometric properties of bounding boxes. In addition, a modified AssemFormer module is employed, using a stacking-decomposition strategy along with residual and skip connections to facilitate effective fusion of local and global features. This enhances the network’s ability to detect small foreign objects in cluttered scenes. Furthermore, a Rectangular Self-Calibrated Module [13] is designed to extract global directional information through horizontal and vertical pooling, construct adaptive attention regions, and apply Large-Kernel Stripe Convolution for contour-aware calibration. This allows attention regions to better align with the true shapes of target objects. This paper also introduces feature fusion and refinement mechanisms, combining Batch Normalization [14] and MLP (Multilayer Perceptron) [15] to enrich deep semantic features and improve boundary segmentation and category discrimination in complex scenes. In order to further improve the generalization performance, the model uses a five-fold cross-validation [16] strategy for training.

The improved YOLOv10-based network proposed in this study addresses key challenges in Pu-erh tea foreign object detection. It provides a practical and efficient solution for quality control in tea production and offers theoretical and methodological references for advancing intelligent foreign object detection in the food industry, laying a solid foundation for the intelligence and automation-based transformation of tea production.

## 2. Materials and Methods

### 2.1. Image Acquisition and Dataset Construction

The image data used in this study were collected at the Laobanzhang tea plantation base in Xishuangbanna Prefecture, Yunnan Province, located in southwestern China (100° E, 21° N). This region features a typical tropical monsoon climate, with an average annual temperature of approximately 22 °C, a cumulative effective temperature (≥10 °C) of 8761 °C, and an average annual relative humidity of 79%. As a high-quality production area for Pu-erh tea, it is representative and well-suited for dataset construction.

In order to construct a high-quality image dataset suitable for intelligent detection of foreign bodies in Pu-erh tea, a Canon EOS R5 camera (Canon (China) Limited, Beijing, China) equipped with an RF24-105 mm lens was used as an image acquisition device to ensure the high resolution and clarity of the image. To improve the model’s detection performance under practical operating conditions, the experimental setup emulated the working distance between industrial sorting equipment and tea leaves in actual production lines, with imaging distance being maintained between 10 and 20 cm. To ensure uniform exposure and complete capture of image details, the fixed sensitivity (ISO) was set to 200, the aperture value to f/4.0, and the shutter speed to 1/300 s. To enhance the model’s adaptability and robustness across diverse real-world scenarios and improve its generalization capability for detecting foreign objects in Pu-erh tea, we proposed a multi-illumination acquisition strategy during the data collection phase. This approach simulated various interference conditions that may occur in actual tea production environments, thereby ensuring reliable detection performance under complex operational conditions.

In this study, a total of 1209 original image samples were collected, and nearly 1700 labels were obtained after manual labeling, covering bamboo leaves, pebbles, tree leaves, and hair. All image samples were labeled by the Make Sense tool, and a visualization of the labeling results is shown in Figure 1. Figure 1A shows the number distribution histogram of various foreign bodies in the data set. Figure 1B reflects the distribution characteristics of different types of foreign objects in length and width by uniformly placing the center points of all foreign object labels in the center of the image. Figure 1C shows the spatial distribution of all foreign object labels in the image. Figure 1D shows the width–height ratio distribution of all foreign object labeling boxes. Finally, Figure 1E shows the detailed distribution characteristics of foreign object labels in the original dataset.

### 2.2. Data Augmentation

To improve the model’s generalization performance [17,18] in complex environments with varying illumination conditions while preventing overfitting and enhancing robustness to multi-angle viewing scenarios, this study implemented a comprehensive image augmentation strategy to increase sample diversity and enrich feature representation capability. As shown in Figure 2, geometric transformation enhancement, random rotation, scaling, cropping and flipping were used to improve the robustness of the model to foreign objects in spatial position and direction changes. A color space enhancement strategy was introduced to enhance the performance of the model under various illumination and color conditions by randomly perturbing parameters such as the brightness, contrast, saturation, and hue of the image. Based on the HSV space [19] enhancement, the hue, saturation, and brightness were disturbed to simulate the phenotypic characteristics of tea foreign bodies under different conditions. Mosaic data enhancement was introduced to break the spatial structure limitation of the original image and enhance the detection ability of the model in multi-target mixed and partial occlusion situations.

### 2.3. YOLOv10 Network Improvements

As one of the new generation of algorithms in the field of target detection, the YOLOv10 network [20,21] delivers high detection speed and accuracy. However, in the face of complex background interference and small-size foreign object targets, low-light, multi-angle and other actual scenes, YOLOv10 still has problems such as insufficient feature extraction and inadequate target positioning accuracy.

To address the above issues, this study made improvements to the YOLOv10 network [22,23]. The architecture of the improved YOLOv10 network is shown in Figure 3, and its detailed parameters are provided in Table 1 for reference and reproducibility. In the table, From indicates the input layer connected to this module, −1 means the input is the output of the previous module, Module refers to the name of the module, and Arguments represent the specific parameters of this module.

#### 2.3.1. AssemFormer Optimization

Foreign objects in Pu-erh tea are often small and visually similar to tea leaves; this makes them difficult to distinguish, leading to degraded detection accuracy in the baseline YOLOv10 network. To address this, we optimized YOLOv10 by integrating AssemFormer, a hybrid module that combines the local feature extraction strengths of CNNs (Convolutional Neural Networks) [24] with the global context modeling capabilities of ViTs (Vision Transformers) [25]. This design effectively preserves fine-grained features and mitigates information loss for small targets. In this framework, convolutional layers efficiently capture local visual patterns such as edges, corners, and textures, which are essential for precise object localization. The Vision Transformer, based on a self-attention mechanism, models long-range dependencies by dividing the image into non-overlapping patches and transforming them into tokenized embeddings.

The overall processing pipeline is shown in Figure 4. Input images are first processed by CNN layers to extract local features and generate structured feature maps. These maps are partitioned into patches and flattened into embeddings suitable for transformer input. To integrate both local and global information, a stacking–unstacking mechanism is employed. In the stacking phase, feature maps are aggregated into higher-dimensional tensors, enriching the global context available to the transformer. Following transformer processing, the unstacking phase restores the original spatial structure, allowing continued processing through downstream convolutional layers.

To avoid information loss and ensure the stability of the training process, Residual Connections and Skip Connections are used. A residual connection allows the input to be transmitted directly to the next layer through a jump connection, which is added to the output of the current layer to prevent the gradient from disappearing. Skip connections enable direct information propagation across network layers, facilitating effective fusion of multi-scale feature representations. This architectural design ensures optimal utilization of hierarchical feature information during Pu-erh tea foreign object detection, thereby significantly enhancing the model’s accuracy and robustness.

#### 2.3.2. Rectangular Self-Calibrated Module Optimization

As an efficient lightweight target detection network, the YOLOv10 network is widely used in real-time target detection tasks due to its low computational burden and fast inference speed. However, it still has some limitations in feature representation, especially in boundary modeling [26,27] and foreground object classification. It often has difficulty in accurately dealing with object boundaries in complex scenes, resulting in inaccurate boundary segmentation and classification errors. In view of the above problems, this study used Rectangular Self-Calibrated Module optimization to improve the performance of the network in boundary modeling and foreground classification by enhancing the position modeling ability of foreground objects. The rectangular self-calibration module is composed of several key components. Its core is Rectangular Self-Calibrated Attention, BN (batch normalization) [28,29], MLP, and Residual Connection are also introduced.

Rectangular Self-Calibrated Attention mainly extracts the horizontal and vertical global context information of the image through horizontal pooling and vertical pooling, respectively, to capture the axial global relationship of the object in space and generate axial vectors in two directions. By adding these two axial vectors together, a rectangular attention region can be formed to accurately model the foreground object, as shown in Equations (1) and (2). Equation (1) represents the Shape Self-Calibration Function, which is mainly used to adjust the shape of the rectangular attention region to make it closer to the actual contour of the foreground object. In the equation, ψ denotes the large-core fringe convolution operation, k denotes the size of the convolution kernel, ϕ represents batch normalization, and δ represents activation function.(1)$$\xi_{c}\left(\bar{y}\right)=\delta (\psi_{k\times 1}(\phi \left(\psi_{1\times k}(\bar{y})))\right)$$(2)$$\xi_{F}\left(x,y\right)=\psi_{3\times 3}\left(x\right)⊙y$$

Equation (2) represents feature fusion, which aims to enhance the feature representation of foreground objects. This process strengthens attention to foreground objects by fusing the attention features adjusted by Rectangular Self-Calibrated Attention with the input features. In the equation, $\psi_{3\times 3}$ denotes a 3 × 3 deep convolution kernel and ⊙ denotes the Hadamard Product.

BN and MLP jointly enhance the robustness and nonlinear expressiveness of learned features. BN standardizes each layer’s output to zero mean and unit variance, accelerating training and reducing internal covariate shift. The MLP, consisting of multiple fully connected layers and activation functions, enables modeling of more complex feature relationships. To preserve original feature information and enable gradient flow, a residual connection adds the module’s output back to the input. Equation (3) summarizes the complete output of the module. In the equation, $H_{P}\left(x\right)$ and $V_{P}(x)$ are horizontal and vertical pooling operations, ⊕ denotes element-wise addition, and ρ represents BN and MLP processing.(3)$$F_{out}=\rho (\xi_{F}(x,\xi_{c}\left(H_{P}\left(x\right)⊕V_{P}(x)))\right)+x$$

#### 2.3.3. Loss Function Optimization

In Pu-erh tea foreign object detection, the YOLOv10 network often exhibits localization inaccuracies when dealing with objects of varying sizes and aspect ratios. This is primarily due to the limitations of conventional IoU-based loss functions, which fail to capture fine-grained geometric discrepancies between bounding boxes.

To address this challenge, we introduce the MPDIoU loss function [30,31], which enhances the original IoU formulation by incorporating a minimal corner-point distance metric. This allows for a more precise assessment of the geometric alignment between predicted and ground-truth boxes. The complete formulation is shown in Equations (4) through (8). Let $\left(x_{prd1\;},\;y_{prd1}\right)$ and $\left(x_{gt2},{\;y}_{gt2}\right)$ represent the top-left and bottom-right coordinates of the predicted bounding box, and $\left(x_{gt1},{\;y}_{gt1}\right)$ and $\left(x_{gt2},{\;y}_{gt2}\right)$ denote the corresponding coordinates of the ground-truth box. The areas of the ground-truth and predicted boxes are denoted by $A_{gt}$ and $A_{prd}$, respectively. The corner distances $d_{1}$ and $d_{2}$ measure the Euclidean distance between the top-left and bottom-right corners of the prediction and ground-truth boxes. The values w and h are the width and height, respectively, of the minimal enclosing box.(4)$$IoU=\frac{A_{gt}\cap A_{prd}}{A_{gt}\cup A_{prd}}$$(5)$$MPDIoU=IoU-\frac{d_{1}^{2}}{w^{2}+h^{2}}-\frac{d_{2}^{2}}{w^{2}+h^{2}}$$(6)$$d_{1}^{2}={\left(x_{prd1}-x_{gt1}\right)}^{2}+{\left(y_{prd1}-y_{gt1}\right)}^{2}$$(7)$$d_{2}^{2}={\left(x_{prd2}-x_{gt2}\right)}^{2}+{\left(y_{prd2}-y_{gt2}\right)}^{2}$$(8)$$L_{MPDIoU}=1-MPDIoU$$

### 2.4. Five-Fold Cross-Validation

To improve the generalization capability of the YOLOv10 network for Pu-erh tea foreign matter detection task, five-fold cross-validation [32] was used to optimize the network training. As a commonly used statistical method, five-fold cross-validation aims to provide a reliable estimate of model performance through repeated training and verification, and effectively reduce the bias caused by the randomness of data partitioning.

As illustrated in Figure 5, the original dataset was randomly partitioned into five mutually exclusive subsets of equal size. In each validation cycle, one subset served as the validation set while the remaining four subsets constituted the training set. The network was iteratively trained on the training set and evaluated on the validation set. This five-fold cross-validation approach ensures comprehensive model assessment across diverse data partitions, effectively mitigating overfitting risks while enhancing the model’s generalization capability. Consequently, the proposed method demonstrated improved accuracy and robustness in Pu-erh tea foreign object detection.

### 2.5. Evaluation Metrics

In order to further assess the performance of the improved YOLOv10 network in the Pu-erh tea foreign matter detection task, this study introduced Precision, Recall, F1, AP (Average Precision) [33] and mAP (mean Average Precision) [34] as the performance evaluation indicators of the model. Precision quantifies the ratio of correctly identified foreign objects to all predicted foreign objects, while recall measures the proportion of successfully detected foreign objects relative to all actual foreign objects present. The F1-score provides a comprehensive evaluation by combining both precision and recall metrics. This is formulated in Equations (9)–(11), where TP denotes true positives (correctly detected foreign objects), FP represents false positives (misidentified foreign objects), and FN indicates false negatives (undetected foreign objects). Average Precision corresponds to the area under the precision-recall curve across varying Intersection over Union thresholds, with recall and precision plotted on the horizontal and vertical axes, respectively. The mean Average Precision, calculated as the mean of AP values across all categories, serves as a robust indicator of the model’s overall performance in Pu-erh tea foreign object detection tasks.(9)$$Precision=\frac{T_{P}}{T_{P}+F_{P}}$$(10)$$Recall=\frac{T_{P}}{T_{P}+F_{N}}$$(11)$$F1=2*\frac{Precision*Recall}{Precision+Recall}$$(12)$$AP=\sum_{i=1}^{n-1}\left(r_{i+1}-r_{i})P_{inter}(r_{i}+1\right)$$(13)$$mAP=\frac{\sum_{i=1}^{k}{AP}_{i}}{k}$$

## 3. Results

To evaluate the performance of the proposed CV-YOLOv10-AR-M network in the task of detecting foreign matter in Pu-erh tea, five target detection models were used for comparative experiments. These were CV-YOLOv10-AR-M (the enhanced version of YOLOv10 included AssemFormer (A), Rectangular Self-Calibrated Module (R), MPDIoU (M), and five-fold cross-validation (CV)), the original YOLOv10, YOLOv12, a grouping corner detection network, and RT-DETR,. All models were trained in the same hardware and software environment. The training was performed on a system equipped with an NVIDIA GeForce RTX 4060Ti (Colorful Ultra W OC, 16 GB) GPU and an Intel Core i5-12600F CPU, running Windows 10 with GPU driver version NVIDIA-SMI 561.09. The network development environment was Python 3.9 and PyCharm 2024. The batch size was set to 128, and the training proceeded for 300 epochs. The optimizer used for training the YOLOv10 network was SGD, with an initial learning rate of 0.01. The training was stopped after 50 epochs without significant improvement in the model. The number of data loading working threads was eight.

### 3.1. Model Performance Analysis

In the task of Pu-erh tea foreign object detection, loss functions are key indicators of optimization progress, as they quantify the deviation between predicted outputs and ground-truth labels. As training advances and the loss values approach their minima, the model’s detection accuracy improves, its generalization ability is enhanced, and the likelihood of both false positives and false negatives decreases. To solve the potential overfitting problem when the network detects small targets in complex scenes, this paper analyzed the loss curves of CV-YOLOv10-AR-M and YOLOv10 networks in detail. As illustrated in Figure 6, the CV-YOLOv10-AR-M and YOLOv10 models start with similar Box and Dfl (Distribution Focal Loss) [35] values, while the initial Cls (Classification) loss of CV-YOLOv10-AR-M is slightly higher. During training, however, the CV-YOLOv10-AR-M network exhibits a significantly faster loss reduction rate. By the 30th epoch, all three loss components in CV-YOLOv10-AR-M have dropped below those of the baseline YOLOv10 model and continue outperforming the baseline through the remainder of training. In the One-to-Many Head task, the CV-YOLOv10-AR-M model achieves final Box, Cls, and Dfl loss values of approximately 1.17, 0.61, and 1.07, respectively, representing reductions of 14.60%, 19.74%, and 20.15%, compared with YOLOv10′s corresponding values of 1.37, 0.76, and 1.34. In the One-to-One Head task, the loss values stabilize around 1.29, 0.56, and 1.07, demonstrating improvements of 10.42%, 29.11%, and 20.15%, respectively, over the original model. Although the improved YOLOv10 network demonstrates significant advantages in external object detection tasks, there are still some false positives and missed detections. The research results indicate that these shortcomings are due to the high similarity in shape and color between certain tea leaves and foreign objects.

As shown in Figure 7, different color thin lines express different categories, and dark blue thick lines are all categories. The CV-YOLOv10-AR-M network achieves Precision of 93.61%, Recall of 94.71%, and an F1-score of 94.16%. These represent improvements of 5.35%, 11.72%, and 8.61%, respectively, over the baseline YOLOv10 model, demonstrating that the enhanced network achieves superior performance in both localization precision and object classification accuracy. The performance improvement provides a reliable technical foundation for intelligent production and quality assurance in the Pu-erh tea industry.

### 3.2. Ablation Study

To verify the effectiveness of the YOLOv10-AR-M network in Pu-erh tea foreign object detection and assess the individual contributions of the AssemFormer, Rectangular Self-Calibrated Module, and MPDIoU enhancements, we conducted a set of ablation experiments. The results are summarized in Table 2. The AssemFormer module improved the YOLOv10 network by 0.30% in Precision, 3.53% in Recall, and 2.36% in mAP, enhancing the model’s perception of small targets in complex scenes. The Rectangular Self-Calibrated Module contributed improvements of 0.65% in Precision, 5.26% in Recall, and 3.17% in mAP, primarily boosting boundary localization accuracy. MPDIoU optimization increased Precision by 1.39%, Recall by 4.39%, and mAP by 1.98%, significantly enhancing localization precision. When all three components were integrated, the YOLOv10-AR-M network achieved an overall improvement of 5.35% in Precision, 11.72% in Recall, and 8.32% in mAP, compared with the original YOLOv10, thus demonstrating more accurate and robust foreign object detection.

To further illustrate the contributions of the AssemFormer, Rectangular Self-Calibrated Module, and MPDIoU components, we visualized model attention using Grad-CAM (Gradient-weighted Class Activation Mapping) [36], as shown in Figure 8. Grad-CAM is a widely used visualization method for CNNs that generates heatmaps highlighting the spatial regions most influential in the decision-making of models. In this task, Pu-erh tea images [37] were passed through the trained YOLOv10 model, and class-specific gradients were backpropagated to the final convolutional feature maps. These gradients were globally averaged across channels to compute importance weights, which were then used to weight and sum the feature maps, producing the final attention heatmap. As shown in Figure 8, the YOLOv10-AR-M network demonstrates enhanced focus on small-scale objects [38], cluttered backgrounds, and fine-grained details. It exhibits more precise attention regions, particularly when detecting tiny foreign objects embedded within tea leaves, validating the effectiveness of the proposed architectural enhancements.

### 3.3. Comparative Model Evaluation

To comprehensively evaluate the detection performance of the improved CV-YOLOv10-AR-M network in the detection of foreign bodies in Pu-erh tea, a comparative experiment involving five target detection models was carried out. These included CV-YOLOv10-AR-M, the original YOLOv10, YOLOv12, CornerNet, and RT-DETR. The results are summarized in Table 3. Compared with YOLOv10, the CV-YOLOv10-AR-M model achieved improvements of 5.35% in Precision, 11.72% in Recall, 8.62% in F1-score, and 8.32% in mAP. Against YOLOv12, the improvements were 8.45%, 11.81%, 10.15%, and 8.49%. Relative to CornerNet, the respective gains were 15.20%, 20.59%, 17.96%, and 16.03%. Against RT-DETR, the improvements were 8.47%, 15.69%, 12.19%, and 11.19%.

To further verify the robustness and adaptability of CV-YOLOv10-AR-M, additional comparative evaluations were performed under challenging conditions, including small targets, cluttered backgrounds, low-light environments, and blurred images. Representative detection results are shown in Figure 9. To ensure the objectivity of the evaluation, we used an independent external dataset which was collected from the Hekai tea base in Xishuangbanna, Yunnan Province. Experimental findings confirmed that CV-YOLOv10-AR-M provided significantly enhanced detection performance over the baseline YOLOv10, particularly in difficult visual scenarios. It substantially reduced both false positives and false negatives, offering a robust and deployable solution for intelligent quality control in the Pu-erh tea industry.

## 4. Conclusions

This study proposes the CV-YOLOv10-AR-M network, a multi-module enhanced model that significantly improves the accuracy and robustness of Pu-erh tea foreign object detection. By integrating MPDIoU loss, AssemFormer, Rectangular Self-Calibrated Module, and five-fold cross-validation into the YOLOv10 architecture, the model addresses challenges such as small-object localization, complex backgrounds, and generalization in real-world settings. Key findings are as follows:

(1) Model training results show that CV-YOLOv10-AR-M achieved faster loss convergence and consistently lower loss values than the original YOLOv10. At epoch 30 and beyond, Box, Cls, and Dfl losses remained below those of the baseline. In the One-to-Many Head task, the model achieved final losses of approximately 1.17, 0.61, and 1.07, reducing loss values by 14.60%, 19.74%, and 20.15%, respectively. In the One-to-One Head task, final losses of 1.29, 0.56, and 1.07 reflected respective improvements of 10.42%, 29.11%, and 20.15%.

(2) Ablation experiments showed that each module contributed meaningfully to overall performance. AssemFormer improved Precision, Recall, and mAP by 0.30%, 3.53%, and 2.36%, respectively. The Rectangular Self-Calibrated Module improved them by 0.65%, 5.26%, and 3.17%, respectively, and MPDIoU by 1.39%, 4.39%, and 1.98%, respectively. Combined, these enhancements led to total improvements of 5.35%, 11.72%, and 8.32% in Precision, Recall, and mAP, respectively, demonstrating the significant complementary effects of the individual modules.

(3) In comparative testing, CV-YOLOv10-AR-M outperformed YOLOv10, YOLOv12, CornerNet, and RT-DETR in all key metrics. Gains in Precision were 5.35%, 8.45%, 15.20%, and 8.47%, respectively; in Recall, 11.72%, 11.81%, 20.59%, and 15.69%, respectively; in F1-score, 8.62%, 10.15%, 17.96%, and 12.19%, respectively; and in mAP, 8.32%, 8.49%, 16.03%, and 11.19%, respectively.

The CV-YOLOv10-AR-M network demonstrates strong generalization capabilities and sets a robust technical foundation for scalable, intelligent quality control in Pu-erh tea production. Its adaptability to challenging visual environments and improved performance across all key metrics position it as a practical and deployable solution in the agricultural and food safety sectors.

## Figures and Tables

**Figure 1 foods-14-01680-f001:**
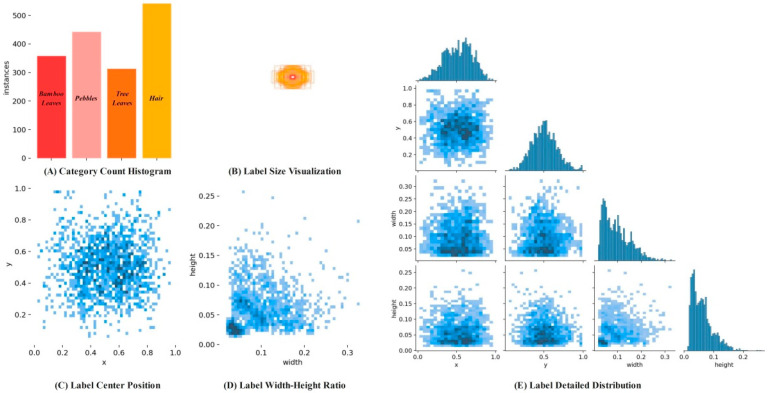
Visualization of dataset label parameters.

**Figure 2 foods-14-01680-f002:**
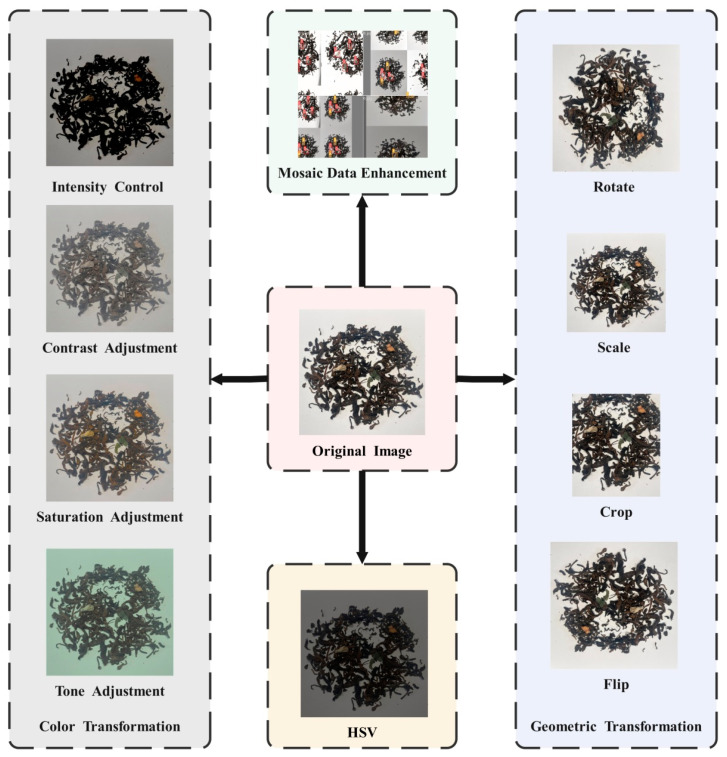
Data augmentation to expand the dataset.

**Figure 3 foods-14-01680-f003:**
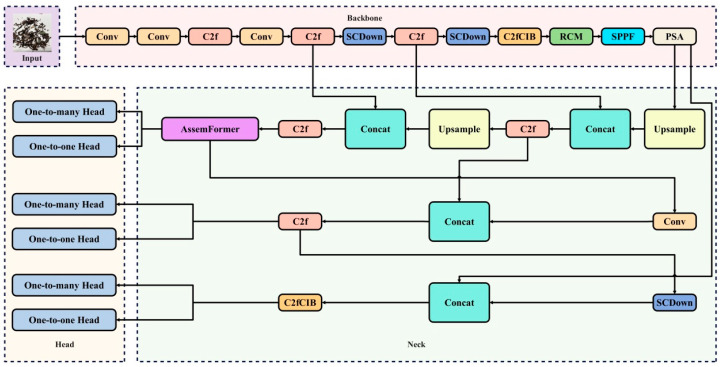
Architecture of the improved YOLOv10 network.

**Figure 4 foods-14-01680-f004:**
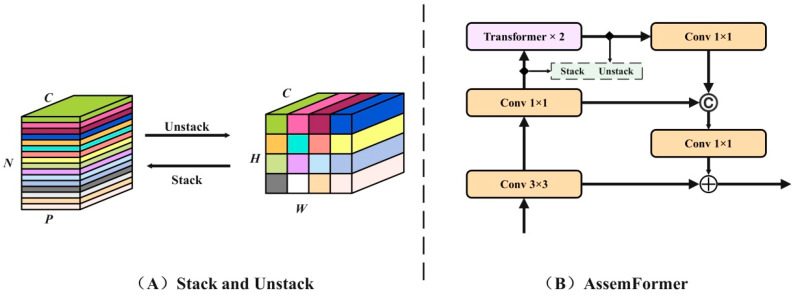
Implementation process of AssemFormer.

**Figure 5 foods-14-01680-f005:**
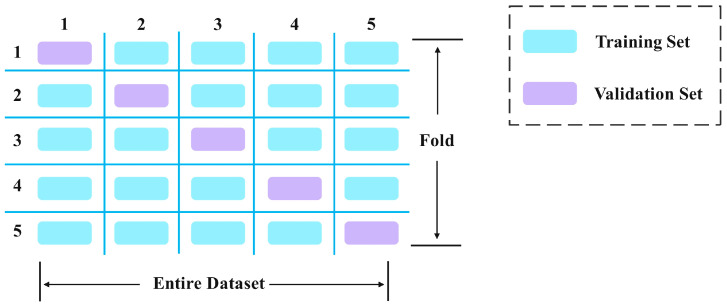
Illustration of five-fold cross-validation.

**Figure 6 foods-14-01680-f006:**
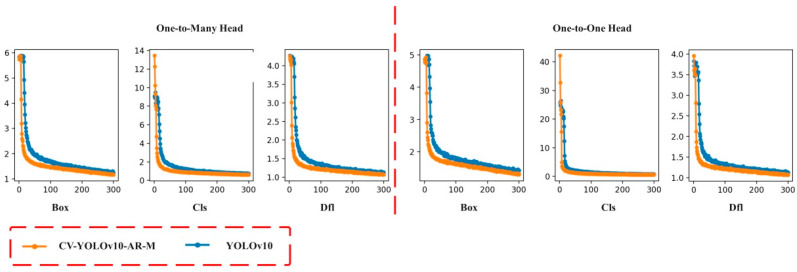
Loss function curves during the training of YOLOv10 and CV-YOLOv10-AR-M networks.

**Figure 7 foods-14-01680-f007:**
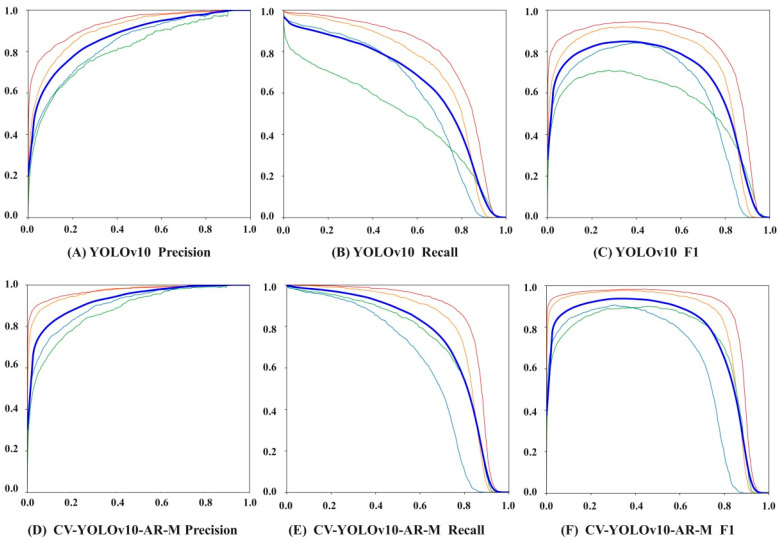
Precision, Recall, and F1-scores of YOLOv10 and CV-YOLOv10-AR-M networks.

**Figure 8 foods-14-01680-f008:**
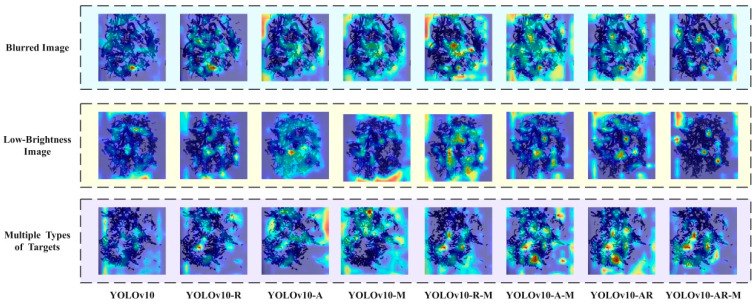
Grad-CAM heatmaps for Pu-erh tea foreign object detection.

**Figure 9 foods-14-01680-f009:**
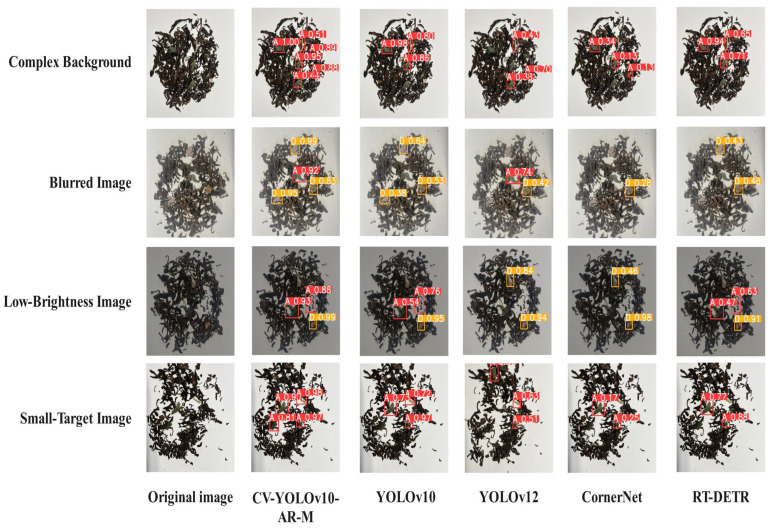
External validation results of the model.

**Table 1 foods-14-01680-t001:** Detailed parameters of the improved YOLOv10 network.

ID	From	Module	Arguments
0	−1	Conv	[3, 16, 3, 2]
1	−1	Conv	[16, 32, 3, 1]
2	−1	C2f	[32, 32, 1, True]
3	−1	Conv	[32, 64, 3, 2]
4	−1	C2f	[64, 64, 2, True]
5	−1	SCDown	[64, 128, 3, 2]
6	−1	C2f	[128, 128, 2, True]
7	−1	SCDown	[128, 256, 3, 2]
8	−1	C2fCIB	[256, 256, 1, True, True]
9	−1	RCM	[256]
10	−1	SPPF	[256, 256, 5]
11	−1	PSA	[256, 256]
12	−1	Upsample	[None, 2, ‘nearest’]
13	[−1, 6]	Concat	[1]
14	−1	C2f	[384, 128, 1]
15	−1	Upsample	[None, 2, ‘nearest’]
16	[−1, 4]	Concat	[1]
17	−1	C2f	[192, 64, 1]
18	−1	AssemFormer	[64]
19	−1	Conv	[64, 64, 3, 2]
20	[−1, 14]	Concat	[1]
21	−1	C2f	[192, 128, 1]
22	−1	SCDown	[128, 128, 3, 2]
23	[−1, 11]	Concat	[1]
24	−1	C2fCIB	[384, 256, 1, True, True]
25	[18, 21, 24]	v10Detect	[4, 64, 128, 256]

**Table 2 foods-14-01680-t002:** Results of ablation study on the YOLOv10-AR-M network.

Model	Precision (%)	Recall (%)	mAP (%)	Layers	Parameters	Gradients
YOLOv10	88.26	82.99	89.15	402	2,496,998	2,496,982
YOLOv10-R	88.91	88.25	92.32	421	2,769,894	2,769,878
YOLOv10-A	88.56	86.52	91.51	464	2,555,368	2,555,352
YOLOv10-M	86.87	87.38	91.13	402	2,496,998	2,496,982
YOLOv10-R-M	90.33	91.12	94.77	421	2,769,894	2,769,878
YOLOv10-A-M	90.20	90.62	94.42	464	2,555,368	2,555,352
YOLOv10-AR	92.07	92.36	95.91	483	2,828,264	2,828,248
YOLOv10-AR-M	93.61	94.71	97.47	483	2,828,264	2,828,248

**Table 3 foods-14-01680-t003:** Model comparison results.

Model	Precision (%)	Recall (%)	F1 (%)	mAP (%)
CV-YOLOv10-AR-M	93.61	94.71	94.16	97.47
YOLOv10	88.26	82.99	85.54	89.15
YOLOv12	85.16	82.9	84.01	88.98
CornerNet	78.41	74.12	76.2	81.44
RT-DETR	85.14	79.02	81.97	86.28

## Data Availability

The original contributions presented in the study are included in the article, further inquiries can be directed to the corresponding author.

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
