# Peer review of "CV-YOLOv10-AR-M: Foreign Object Detection in Pu-Erh Tea Based on Five-Fold Cross-Validation"

_foods, 2025, doi:10.3390/foods14101680_

Round 1

Reviewer 1 Report

Comments and Suggestions for Authors

Brief summary

The manuscript "CV-YOLOv10-AR-M: Foreign Object Detection in Pu-erh Tea Based on 5-Fold Cross-Validation" introduces an improved detection model based on the YOLOv10 framework for detecting tiny foreign objects in Pu-erh tea under complicated conditions. The model incorporates several architectural advances: MPDIoU loss for better bounding box accuracy, AssemFormer for local-global feature fusion, and Rectangular Self-Calibrated Module for enhanced attention. The data is acquired under real-life production environments, and model performance is demonstrated with ablation studies and comparative evaluations with reasonable improvements over YOLOv10, CornerNet, and RT-DETR.

Major comments

Although reasonable, English in the manuscript needs polishing. Examples:
Line 16: please replace "the YOLOv10 network is optimized, which effectively enhancing." with "the YOLOv10 network is optimized to effectively enhance."
Line 29: "lays a solid foundation for promoting the intelligent transformation of the tea industry" is overused phrase: try a different wording choice. A thorough revision by a fluent or native speaker editor is strongly advised.

Formulations in introduction and methods parts are very generic and textbook-style (e.g., lines 52–58). Please restate these parts with more descriptive references or distinctive wording for better originality.

Lines 82-100: the description of each module (AssemFormer, MPDIoU, etc.) is mentioned in different sections (abstract, introduction, and methods), making the manuscript redundant. Avoid repeating architectural pieces more than once unless absolutely required.

Lines 94-100: although architectural details are mentioned clearly, there is no mention of sharing the code or trained model weights. Please provide the code or weights for full reproducibility, particularly as the method is proposed for industrial practice.

Lines 102-132: the dataset is drawn from all from a single source tea plantation (Laobanzhang, Yunnan). This introduces a risk of geographical or environmental bias, potentially constraining model applicability to other larger-scale production scenarios. Please consider mentioning this limitation explicitly and proposing future plans for outside validation on diverse sources.

Lines 295-347: the manuscript does not include an analysis of potential overfitting during training. While 5-fold cross-validation is used, no learning curves or validation/training loss comparisons are displayed. Adding learning curves or at least a brief overfitting discussion is advised to validate the model's generalization ability.
Although quantitative outcomes are adequately reported, qualitative error analysis (e.g., common false positives or false negatives) is missing from the manuscript. Showing several failure cases or usual misclassifications would assist readers in better comprehending the model's limitations.

Lines 301-305: whilst hardware specs are provided, basic training parameters such as optimizer type, learning rate, scheduler, and early termination criteria are missing from the manuscript. Please add these setup details to improve reproducibility.

The publication ascribes real-time detection potential, yet no inference time or FPS (frames per second) is reported for the model. Please consider adding a runtime analysis (on CPU and/or GPU) to test the model's real-time suitability.

The dataset used is not compared to any existing public benchmarks or made publicly available. This limits the broader relevance and reproducibility of the study. Please clarify if the dataset will be released or if comparisons with open datasets are planned.

Lines 366-375 (Table 3): only three other models are compared. The evaluation will be made stronger by comparing with more YOLO variations (e.g., YOLOv5, YOLOv7) or transformer-based detectors like Deformable DETR.

Tables 2-3: whilst improved performance is explicitly reported, no statistical analysis is given to validate reported differences. It is recommended that you add significance testing (e.g., paired t-test or confidence intervals) to validate comparative claims.

Minor comments

Figure 1, 2, 4, 5, 6, 7, 9: captions are very short or unhelpful. Use longer descriptive captions, particularly for technical figures like network architectures or Grad-CAM maps.

Lines 448-449: the list of abbreviations is useful, though all are not consistently used (e.g., "Dfl" not defined earlier clearly). Please define every such acronym on first use in the main text.
A few references are very recent (2024 - 2025), and this is excellent, though the over-reliance on MDPI and domain-specific journals may be too narrow. Please use a few more machine learning or computer vision sources from larger conferences like CVPR, ICCV, or IEEE T-PAMI.

The baseline models compared with (CornerNet and RT-DETR) are not as widely used as in current industrial or agricultural environments. It would make the study better to compare with more widely used detectors such as YOLOv5 or YOLOv7.

A few references (e.g., refs. [3]–[4]) are quite generic and loosely related to the precise claims they endorse. Please ensure all references have direct relevance and contribute to making a point in the discussion.
Some sentences in the manuscript (e.g., "lays a solid foundation for intelligent transformation") are markedly over-the-top rhetorical and should perhaps be reduced. I will try to use a more objective and succinct scientific writing style in the conclusions and abstract.

Comments on the Quality of English Language

The English language in the manuscript requires significant improvement to enhance clarity, originality, and professionalism.

Several areas need attention, including sentence structure, the use of overly generic phrases, redundancy in the presentation of technical modules, and some over-the-top rhetorical expressions.

Author Response

Thanks very much for your time to review this manuscript. I really appreciate you’re your comments and suggestions. We have considered these comments carefully and tried our best to address every one of them. And the corresponding part in the text has been modified using red font.

Comments 1: though reasonable, English in the manuscript needs polishing. Examples: Line 16: please replace "the YOLOv10 network is optimized, which effectively enhancing." with "the YOLOv10 network is optimized to effectively enhance." Line 29: "lays a solid foundation for promoting the intelligent transformation of the tea industry" is overused phrase: try a different wording choice. A thorough revision by a fluent or native speaker editor is strongly advised.

Response1: Thank you for your valuable feedback on our manuscript. We have carefully read your suggestions and made revisions throughout the entire manuscript.

Comments 2:Formulations in introduction and methods parts are very generic and textbook-style (e.g., lines 52–58). Please restate these parts with more descriptive references or distinctive wording for better originality.

Response 2: Thank you for your valuable feedback on our manuscript. We have carefully reviewed your suggestions and made revisions accordingly. Regarding your point about the introduction and methods sections being too general and textbook-like (e.g., lines 52-58), we have rewritten these parts.

Comments 3:Lines 82-100: the description of each module (AssemFormer, MPDIoU, etc.) is mentioned in different sections (abstract, introduction, and methods), making the manuscript redundant. Avoid repeating architectural pieces more than once unless absolutely required.

Response 3:Thank you for your valuable feedback. Regarding the issue of repetition in lines 82-100, we have made revisions. In the abstract, we briefly introduced the optimization approach of the YOLOv10 network. In the introduction, we presented the background and function of each module for the first time. In the methods section, we have streamlined and removed the redundant descriptions of the modules to avoid unnecessary repetition.

Comments 4:Lines 94-100: although architectural details are mentioned clearly, there is no mention of sharing the code or trained model weights. Please provide the code or weights for full reproducibility, particularly as the method is proposed for industrial practice.

Response 4:Thank you for your valuable feedback on our manuscript. Regarding the lack of explanation about the code and training model weights in lines 94-100, we have sent the relevant code to the editorial office. If you need a copy of the code, please feel free to provide your email, and we will send it to you promptly. Additionally, we have included the information about the training model weights in the third section of the article (Results). If you require more detailed weight parameters, we can take screenshots and send them to you at any time, or directly send you the corresponding .py file.

Comments 5:Lines 102-132: the dataset is drawn from all from a single source tea plantation (Laobanzhang, Yunnan). This introduces a risk of geographical or environmental bias, potentially constraining model applicability to other larger-scale production scenarios. Please consider mentioning this limitation explicitly and proposing future plans for outside validation on diverse sources.

Response 5:Thank you very much for your valuable suggestions. We fully agree with your point regarding the potential geographical or environmental bias due to the single source of the dataset. To address this issue, we have conducted external validation through the He Kai Base in Section 3.3, further validating the performance of our improved network on data from different sources.

Comments 6:Lines 295-347: the manuscript does not include an analysis of potential overfitting during training. While 5-fold cross-validation is used, no learning curves or validation/training loss comparisons are displayed. Adding learning curves or at least a brief overfitting discussion is advised to validate the model's generalization ability. Although quantitative outcomes are adequately reported, qualitative error analysis (e.g., common false positives or false negatives) is missing from the manuscript. Showing several failure cases or usual misclassifications would assist readers in better comprehending the model's limitations.

Response 6:Thank you for your valuable feedback on our manuscript. Regarding the overfitting issue and error analysis that you pointed out, we have addressed this in Section 3.1. Since this study involved extensive data augmentation, the loss function curve in Figure 6 clearly shows that overfitting did not occur in this study. Regarding error analysis, although the improved YOLOv10 network demonstrates significant advantages in external object detection tasks, there are still some false positives and missed detections. The research results indicate that this is due to the high similarity in shape and color between certain tea leaves and foreign objects.

Comments 7:Lines 301-305: whilst hardware specs are provided, basic training parameters such as optimizer type, learning rate, scheduler, and early termination criteria are missing from the manuscript. Please add these setup details to improve reproducibility. The publication ascribes real-time detection potential, yet no inference time or FPS (frames per second) is reported for the model. Please consider adding a runtime analysis (on CPU and/or GPU) to test the model's real-time suitability. The dataset used is not compared to any existing public benchmarks or made publicly available. This limits the broader relevance and reproducibility of the study. Please clarify if the dataset will be released or if comparisons with open datasets are planned.

Response 7:Thank you for your valuable feedback on our manuscript. We have made detailed revisions in response to the issues you raised and have added relevant details on training, real-time detection analysis, and the dataset's availability. We have included detailed information about the optimizer type, learning rate, learning rate scheduler, and early stopping criteria used during training in the manuscript. The optimizer used for training the YOLOv10 network is SGD, with an initial learning rate of 0.01. The training was stopped after 50 epochs without significant improvement in the model. The number of data loading worker threads is 8. Regarding the FPS and inference time issue you mentioned, our tests indicate that FPS values may have some degree of variability across different devices, and therefore may not fully reflect the real-time detection capabilities of the model. To better evaluate the model's real-time performance, this study further investigates the real-time detection potential of the model in Section 3.2 through the Layers, Parameters, and Gradients parameters. Regarding your suggestion about the generalizability of the study, in Section 3.3, we used data from the He Kai Base for external validation experiments. As for the dataset availability, future readers can apply for access to the dataset from our research center, and we will provide the relevant data through the official email of the research center.

Comments 8:Lines 366-375 (Table 3): only three other models are compared. The evaluation will be made stronger by comparing with more YOLO variations (e.g., YOLOv5, YOLOv7) or transformer-based detectors like Deformable DETR.

Response 8: Thank you for your valuable feedback on our manuscript. We have made revisions based on your suggestions and included the latest YOLOv12 network for comparison in both the model comparison experiment and the external validation experiment.

Comments 9:Tables 2-3: whilst improved performance is explicitly reported, no statistical analysis is given to validate reported differences. It is recommended that you add significance testing (e.g., paired t-test or confidence intervals) to validate comparative claims.

Response 9:Thank you for your valuable feedback. The performance validation of visual recognition networks typically does not rely on statistical analysis, but instead uses metrics such as Precision, Recall, and mean Average Precision to evaluate the accuracy and effectiveness of the model. Visual recognition is a direct prediction task, and the core questions it addresses are: Can the model correctly identify the target? How many targets were missed? How many were misclassified? In contrast, traditional statistical analysis is often used to compare differences in small sample datasets, with the goal of inferring whether certain phenomena are due to randomness. Additionally, image data is high-dimensional and exhibits nonlinear relationships, with strong spatial dependencies between samples, making it difficult to meet the assumptions required by traditional statistical methods.

Comments 10:Figure 1, 2, 4, 5, 6, 7, 9: captions are very short or unhelpful. Use longer descriptive captions, particularly for technical figures like network architectures or Grad-CAM maps.

Response 10:Have been changed in accordance with the advice given.

Comments 11:Lines 448-449: the list of abbreviations is useful, though all are not consistently used (e.g., "Dfl" not defined earlier clearly). Please define every such acronym on first use in the main text. A few references are very recent (2024 - 2025), and this is excellent, though the over-reliance on MDPI and domain-specific journals may be too narrow. Please use a few more machine learning or computer vision sources from larger conferences like CVPR, ICCV, or IEEE T-PAMI.

Response 11: Have been changed in accordance with the advice given.

Comments 12:The baseline models compared with (CornerNet and RT-DETR) are not as widely used as in current industrial or agricultural environments. It would make the study better to compare with more widely used detectors such as YOLOv5 or YOLOv7.

Response 12: Thank you for your valuable feedback. Regarding your point that baseline models such as CornerNet and RT-DETR are not as widely used in current industrial or agricultural environments compared to other models, we understand your concern. In fact, the YOLOv10 network itself is an optimization and improvement built upon earlier versions like YOLOv5 and YOLOv7. Therefore, to further validate the advantages of our improved model, we have additionally included the YOLOv12 network for comparison in our study. By introducing YOLOv12, we aim to demonstrate the advantages of our model in comparison to the latest YOLO variants, thereby enhancing the credibility and application potential of our research.

Comments 13:A few references (e.g., refs. [3]–[4]) are quite generic and loosely related to the precise claims they endorse. Please ensure all references have direct relevance and contribute to making a point in the discussion. Some sentences in the manuscript (e.g., "lays a solid foundation for intelligent transformation") are markedly over-the-top rhetorical and should perhaps be reduced. I will try to use a more objective and succinct scientific writing style in the conclusions and abstract

Response 13:Have been changed in accordance with the advice given.

Reviewer 2 Report

Comments and Suggestions for Authors

The paper presents a promising approach, CV-YOLOv10-AR-M: Foreign Object Detection in Pu-erh Tea 2 Based on 5-Fold Cross-Validation.

Authors are suggested to clarify the following comments:

  1. The contribution points and novelty of the work are missing at the end of the introduction section.
  2. Figure 3. Architecture of the improved YOLOv10 network (Neck and output are not clear )
  3. Table 1. Detailed parameters of the improved YOLOv10 network ( what is from parameter and -1 means here. )
  4. Figure 4. AssemFormer  (detailed description required)

  5. 5-fold Cross-Validation (row and column classes need to show in Figure 5)  

Author Response

Thanks very much for your time to review this manuscript. I really appreciate you’re your comments and suggestions. We have considered these comments carefully and tried our best to address every one of them. And the corresponding part in the text has been modified using red font.

Comments 1: The contribution points and novelty of the work are missing at the end of the introduction section.

Response 1: Thank you for your valuable feedback on our manuscript. We have revised and expanded the last two paragraphs of the introduction to clearly outline the main contributions and innovations of this study.

Comments 2: Architecture of the improved YOLOv10 network (Neck and output are not clear )

Response 2:  Thank you for your valuable feedback. Regarding the issue of unclear Neck and output sections in the YOLOv10 network architecture diagram, we have adjusted the image to improve its clarity. Since the original image had a very high resolution, which could lead to slow file loading, we have compressed and optimized the image. If you need to view the original high-resolution image, please feel free to contact us, and we will provide the uncompressed high-definition version.

Comments 3: Table 1. Detailed parameters of the improved YOLOv10 network ( what is from parameter and -1 means here. )

Response 3: Thank you for your valuable feedback on our manuscript. From indicates the input layer connected to this module, -1 means the input is the output of the previous module, Module refers to the name of the module, and Arguments represent the specific parameters of this module.

Comments 4: Figure 4. AssemFormer  (detailed description required)

Response 4: Thank you for your valuable feedback. Regarding the AssemFormer section in Figure 4, we have provided a detailed description of the diagram and added a more comprehensive explanation. First, the CNN layers extract local features such as edges, corners, and textures. The image is then divided into non-overlapping blocks, which are flattened and converted into embedding vectors, processed by a vision transformer to model long-range dependencies and capture global context. To effectively combine local and global information, AssemFormer uses a stacking-unstacking mechanism. In the stacking phase, feature maps are aggregated into higher-dimensional tensors to enhance global context; in the unstacking phase, the original spatial structure is restored, ensuring the model retains the original resolution. The module also introduces residual connections and skip connections to prevent information loss and stabilize the training process, ensuring the effective fusion of multi-scale features.

Comments 5: 5-fold Cross-Validation (row and column classes need to show in Figure 5) 

Response 5: Thank you for your valuable feedback. Have been changed in accordance with the advice given.

Round 2

Reviewer 1 Report

Comments and Suggestions for Authors

Dear Authors,

Thank you for sending in a revised version of your manuscript together with a detailed cover letter addressing the comments.

I have carefully reviewed your revisions and am now very happy to inform you that the changes you have addressed do indeed satisfy all the comments I raised.

The manuscript has been significantly improved and now meets the standards required for publication in Foods Journal.

I commend your efforts in responding thoroughly and thoughtfully to the comments.

Congratulations and good luck on the publishing of your work.

Best regards.
